# Symptomatic Vascular Compression of Brainstem May Be Managed Conservatively

**DOI:** 10.3390/life12081179

**Published:** 2022-08-02

**Authors:** Malik Ghannam, Meaghen Berns, Apameh Salari, Lisa Moore, Kevin Brown

**Affiliations:** 1Department of Neurology, University of Iowa Hospitals & Clinics, 200 Hawkins Dr, Iowa City, IA 52242, USA; 2Department of Neurology, University of Minnesota, Minneapolis, MN 55455, USA; berns127@umn.edu (M.B.); asalari@umn.edu (A.S.); moor1940@umn.edu (L.M.); 3VA Medical Center, Minneapolis, MN 55417, USA; kevin.brown13@va.gov

**Keywords:** vertebral artery, medulla, compression

## Abstract

Medulla compression from vertebral artery abnormality is a very rare occurrence with few cases present in the literature. It has been documented to present with a very wide spectrum of clinical symptomatology ranging from asymptomatic to full hemiplegia. There is currently no treatment algorithm in place to guide clinicians encountering such patients but treatments have historically involved major posterior compartment surgical interventions. This case outlined a patient evaluated for dizziness without any other neurological symptoms or signs, found to have abnormal dilatation, elongation, and tortuosity of the vertebral artery with resultant compression of the medulla oblongata. The patient was managed conservatively after discussion of surgical options. This report outlined an important consideration for management of medullar compression by vertebral artery based on symptom severity with the possibility of postponing surgical or endovascular interventions and opting for conservative management with an anti-platelet regimen, particularly in the short term.

## 1. Introduction

Medulla oblongata (MO) compression is a rare clinical entity with few published reports in the literature. Most cases to date describe MO compression secondary to tumor mass effect; it is far less common to find vascular indentation as the primary etiology [1,2,3,4,5,6,7]. Vascular causes of MO compression such as vessel aneurysm, trigeminal nerve malformation, dolichoectasia of the vertebrobasilar arterial system, and tortuous vertebral artery (VA) brainstem compression have been previously described.

Symptomatic conflict of the vertebral artery with the MO, sometimes named Vertebral Artery Medulla Compression Syndrome (VAMCS), may cause neurologic dysfunction with a highly variable clinical presentation; symptoms such as hemiparesis, dysphagia, headache, vertigo, visual changes, and non-descript poor cognition have been reported. One particular case of VAMCS involved a patient presenting with intractable nausea/vomiting and weight loss with full symptom improvement following microvascular decompression [8,9,10,11,12]. Symptoms have also been shown to occur in isolation with two prior cases of VAMCS presenting as dizziness only [13]. In general, however, medulla compression appears to lack any correlation between the range and extent of symptomatology and the site of MO indentation. Additionally, patient symptoms and radiological findings do not match in many instances, which makes medulla compressing lesions difficult to diagnose [13,14,15,16,17,18,19,20,21,22,23,24]. Here, we described a rare case among the literature of medulla compression presenting as dizziness without other focal neurologic signs.

## 2. Presentation

An 83-year-old man with a past medical history of hypertension, generalized anxiety disorder, and asthma presented to the neurology clinic complaining of worsening episodic dizziness and instability over a 1-year period. These episodes mostly occurred in the morning, immediately after waking from bed, and most often resolved by the afternoon. The only symptom associated with these episodes was mild shakiness in the hands. The patient denied a history of syncopal episodes or falls.

With characteristic symptoms of dizziness after position change and mildly positive orthostatic vital signs, the patient was initially diagnosed with orthostatic hypotension but showed minimal improvement with first-line treatments such as medication review for contributory drugs, increased hydration, and compression stockings. The patient underwent a cardiac workup including ambulatory EKG monitoring which showed brief episodes of supraventricular tachycardia that did not align temporally with his reported episodes of lightheadedness, dizziness, and imbalance.

Physical exam revealed no spontaneous or gaze-evoked nystagmus, normal head-shake test, and a normal head-thrust test. Videonystagmography revealed normal saccade, smooth pursuit, optokinetic nystagmus and bilateral normal and symmetrical caloric responses. The only positive neurologic finding was torsional nystagmus during head positional tests (head hanging positions) and Dix-Hallpike maneuver. The nystagmus observed was not fatigable and not suppressed by fixation. The patient did not report a sense of vertigo during these exams. The patient had normal sensation to touch and pinprick throughout as well as intact cranial nerve and strength testing.

Magnetic resonance imaging (MRI) was requested to rule out central lesions and demonstrated an abnormal course of the VA. The VA was patent and visualized to have a tortuous course along the lateral aspect of the MO, displacing it to the contralateral side. Both the seventh and eighth cranial nerves were unremarkable bilaterally (Figure 1). He was prescribed 81 mg daily aspirin and a generous fluid regimen with a referral for vestibular rehabilitation therapy. He was walking several hours per day still with some symptoms but had no falls and no focal deficits at 1-year follow-up.

## 3. Discussion

Compression of the medulla via the vertebral artery or basilar artery represent rare but known clinical phenomena and are also known to cause a spectrum of clinical presentations. This spectrum ranges from mild nonspecific symptoms such as headache and dizziness to cranial nerve abnormalities, deficits resembling brainstem stroke syndromes, hemiplegia, or even compromise of respiratory drive. Therefore, in the context of a symptomatology that ranges in severity, considerations must be made as to how aggressive treatment should be, ranging from no treatment to symptomatic treatment to even posterior decompression surgery if indicated. A further consideration in such cases is that severity of imaging findings often does not correlate with the severity of clinical symptoms. As described in a recent case series by Mahrous et al., one patient demonstrated severe compression on imaging without little to no correlating symptoms [13]. In our case, the patient had symptoms that were provokable with positional changes (particularly laying down to standing up) and were considered to be orthostatic in nature despite only minimally positive orthostatic vital signs. The patient likely had some orthostatic component worsening his VA compression-related dizziness as his symptoms also appeared to have improved slightly after first-line treatments for orthostasis. The management strategy for VAMCS must be tailored to each particular patient taking into consideration the imaging findings and comorbidities, but most importantly, the severity and chronicity of symptomatology. Previous cases of medulla compression that have resulted in surgical treatment have involved four primary microsurgical procedures: vessel mobilization, vessel section with posterior fossa decompression, autologous material inlay with posterior fossa decompression, and lateral vessel retraction assisted with Gore-Tex. It has been reported that in some cases, surgical decompression leads to significant improvement in patients with debilitating symptoms [1]. Unfortunately, other studies have reported contradictory results to surgical intervention with either a lack of improvement or a very temporary improvement in symptoms after surgery [13]. Secondly, conservative management based on addressing each patient complaint individually has included aspirin, warfarin, dipyridamole, and analgesics [1]. Additionally, patients with central vestibular disorders have also been shown to benefit from vestibular rehabilitation therapy [25,26,27,28]. Accordingly, a vestibular rehabilitation exercise regimen was given to our patient and they ultimately reported significant functional, physical and emotional benefits with decreased frequency and severity of the dizziness attacks.

On how this report expands the literature we would engage with 3 recent reports on VAMCS. Tsutsumi et al. report on 57 cases presented of vertebral artery compression of the medulla were on asymptomatic patients and while comparable from an imaging standpoint to the patient in this report, that report details patients without symptoms while our patient had, for over a year prior to presentation and for which other relevant medical workup had been unrevealing, bother some symptoms [29,30,31]. It is our contention that their may then be value in reporting here that conservative medical management alone (daily antiplatelet regimen in this case) may be useful for other clinicians to know.

Conversely, in a report by Li et al. [32], patients in that report were symptomatic. The authors prudently acknowledge that future work is still needed to clarify the natural history and treatment of this condition and in that respect we are offering a case to the literature where antiplatelet management might be useful in said management.

And finally, the report by Savitz et al. [14], while most patients were not treated aggressively in that report (meaning surgically), other patients were treated with antiplatelet regimens but the authors acknowledge that ‘further studies are needed to estimate preferred management’. In that regard we are adding to the call which those authors placed, for more information regarding clinical management.

## 4. Conclusions

Patients with vertebral artery compression of the medulla can present with a wide variety of symptoms for which there exists no correct treatment algorithm to date. This case outlines a patient who presented to the neurology clinic with dizziness only found to have imaging findings consistent with VAMCS. He was ultimately managed with medications and vestibular rehabilitation therapy only based on the minimal severity of symptoms thus avoiding surgical or endovascular interventions at least in the short term. To that end, the main contribution of this report can be understood to add to the limited but growing literature that conservative medical management such as with antiplatelet regimen can be useful in treating the symptomatic patients with VAMCS.

## Figures and Tables

**Figure 1 life-12-01179-f001:**
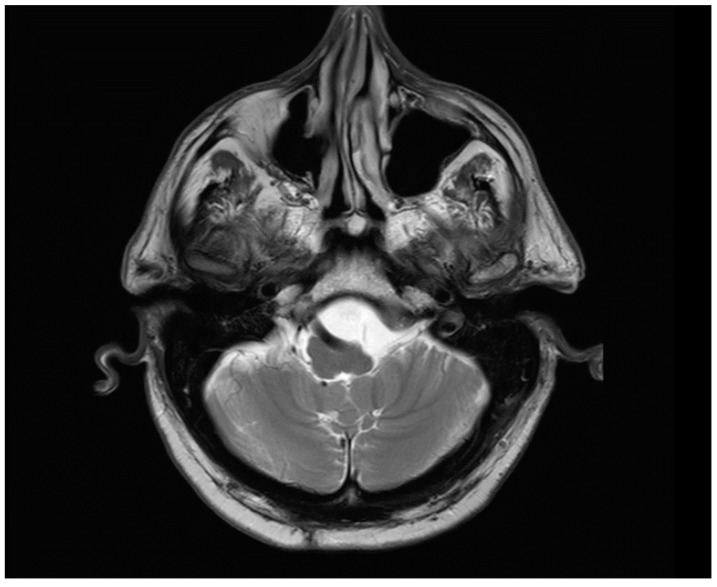
MRI brain demonstrating the left vertebral artery causing chronic indentation on the medulla. Otherwise, mild chronic small vessel ischemic changes are visible. Tiny old lacunar infarct visible in the right cerebellar hemisphere.

## Data Availability

All the data supporting our findings is contained within the manuscript.

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
