# Peer review of "Symptomatic Vascular Compression of Brainstem May Be Managed Conservatively"

_life, 2022, doi:10.3390/life12081179_

Round 1

Reviewer 1 Report

The manuscript "Vascular Compression of Brainstem May Present as Isolated Dizziness and May be Managed Conservatively" by Ghannam and colleagues presents a case of an 83-year-old man with dizziness secondary to a vertebral artery compressing the medulla oblongata. 

There are many things that have to praised about this manuscript:

1. It is very well written.

2. It is very concise and to the point.

3. It is focussed.

It does quit a good job discussing the case, but it fails at properly addressing the substantial question as to what knowledge gap it closes. The Vertebral Artery Medulla Compression Syndrome (VAMCS) is a very rare phenomenon that has not been described extensively in the literature. However, there are some important publications that are not mentioned, but do contain a comparably substantial amount of patients, amongst which some are treated conservatively just as in this case:

* Li et al. 2019 Front Neurol (doi: 10.3389/fneur.2019.01075)

* Tsutsumi et al. 2022 Surg Neurol Int (doi: 10.25259/SNI_1161_2021)

The work by Tsutsumi and colleagues is especially daunting as it shows a staggering 57 cases of asymptomatic compression of the medulla oblongata by the vertebral artery (of which, admittedly, 48 were considered mild, still leaving 9 to have considerable compression). And in the series of Savitz et al. (ref. #4) a total of 6 patients were managed using a conservative approach. So this begs the question: Why should this report of a 83-year-old man be published? What does it add?

I'd suggest elaborating this pivotal question and revise the manuscript accordingly.

Minor comments:

* Abstract: "initial recommendation" seems a little far-fetched

* Introduction: why is "numbness" emphasized in particular?

* Discussion: numbered as fourth section - what happened to section three?

Author Response

The manuscript "Vascular Compression of Brainstem May Present as Isolated Dizziness and May be Managed Conservatively" by Ghannam and colleagues presents a case of an 83-year-old man with dizziness secondary to a vertebral artery compressing the medulla oblongata.

There are many things that have to praised about this manuscript:

  1. It is very well written.
  2. It is very concise and to the point.
  3. It is focussed.

It does quit a good job discussing the case, but it fails at properly addressing the substantial question as to what knowledge gap it closes. The Vertebral Artery Medulla Compression Syndrome (VAMCS) is a very rare phenomenon that has not been described extensively in the literature. However, there are some important publications that are not mentioned, but do contain a comparably substantial amount of patients, amongst which some are treated conservatively just as in this case:

* Li et al. 2019 Front Neurol (doi: 10.3389/fneur.2019.01075)

* Tsutsumi et al. 2022 Surg Neurol Int (doi: 10.25259/SNI_1161_2021)

The work by Tsutsumi and colleagues is especially daunting as it shows a staggering 57 cases of asymptomatic compression of the medulla oblongata by the vertebral artery (of which, admittedly, 48 were considered mild, still leaving 9 to have considerable compression). And in the series of Savitz et al. (ref. #4) a total of 6 patients were managed using a conservative approach. So this begs the question: Why should this report of a 83-year-old man be published? What does it add?

 This is an excellent point made by Reviewer 1 and the Tsutsumi et al report was not aware to the authors at the time of composing this manuscript.  We agree the findings are relevant but also would point out that the 57 cases presented of VAMCS were on asymptomatic patients and while comparable from an imaging standpoint, did not display the symptoms as our patient had done for over a year prior to presentation and for which other relevant medical workup had been unrevealing.  In that sense, there is still value in presenting this information as our patient did improve with medical management (daily aspirin). 

With regard to Li et al, patients in that report were symptomatic.  The authors prudently acknowledge that future work is still needed to clarify the natural history and treatment of this condition and in that respect we are offering a case to the literature where anti-platelet management might be useful in said management.

And with regard to Savitz, while most patients were not treated aggressively in that report (meaning surgically), other patients were treated with anti-platelet regimens but the authors acknowledge that ‘further studies are needed to estimate preferred management’.  In that regard we are adding to the call which those authors placed, for more information regarding clinical management.

To this end, an additional section has been added to the discussion including points made above.

I'd suggest elaborating this pivotal question and revise the manuscript accordingly.

Minor comments:

* Abstract: "initial recommendation" seems a little far-fetched

Thank you for this notification, we have changed the wording accordingly.

* Introduction: why is "numbness" emphasized in particular?

We do not have a clear reason for emphasizing this, and have removed mention of numbness

* Discussion: numbered as fourth section - what happened to section three?

We have addressed this and made the appropriate changes, thank you for noticing.

Reviewer 2 Report

The authors had well written the draft.  I do not have any more questions.

Author Response

Thank you for your review.

Sincerely.

Reviewer 3 Report

Thank you for the opportunity to review your manuscript, Vascular Compression of Brainstem May Present as Isolated Dizziness and May be Managed Conservatively

This Case Report study describing the conservative management of a patient with vertebral artery abnormality resulting in compression of the medulla oblongata.  

The case is well presented, but presents some considerations for improvement.

The authors should review the spacing between paragraphs.

The order of the references used should follow an order of appearance.

In lines 77-78 and 79, there is a text that I consider does not belong in the manuscript. Who should revise it.

The conclusion should not be a final mini-summary of the case presented. It should make clear the contribution of this case to the scientific literature.

Author Response

Thank you for the opportunity to review your manuscript, Vascular Compression of Brainstem May Present as Isolated Dizziness and May be Managed Conservatively

This Case Report study describing the conservative management of a patient with vertebral artery abnormality resulting in compression of the medulla oblongata. 

The case is well presented, but presents some considerations for improvement.

The authors should review the spacing between paragraphs.

The order of the references used should follow an order of appearance.

Thank you for noticing this, we have made the appropriate corrections.

In lines 77-78 and 79, there is a text that I consider does not belong in the manuscript. Who should revise it.

Thank you for noticing this, which was a hangover from the Life formatting process, we have corrected it.

The conclusion should not be a final mini-summary of the case presented. It should make clear the contribution of this case to the scientific literature.

We agree and have made the appropriate changes in the text, thank you for the notification.

Round 2

Reviewer 3 Report

The authors have made the necessary changes. The article is acceptable for publication